# Endocrine-Disrupting Chemicals and Insulin Resistance in Children

**DOI:** 10.3390/biomedicines8060137

**Published:** 2020-05-28

**Authors:** Eleonora Rotondo, Francesco Chiarelli

**Affiliations:** Department of Pediatrics, University of Chieti, I-66100 Chieti, Italy; chiarelli@unich.it

**Keywords:** endocrine-disrupting chemicals, insulin resistance

## Abstract

The purpose of this article is to review the evidence linking background exposure to endocrine-disrupting chemicals (EDCs) with insulin resistance in children. Although evidence in children is scarce since very few prospective studies exist even in adults, evidence that EDCs might be involved in the development of insulin resistance and related diseases such as obesity and diabetes is accumulating. We reviewed the literature on both cross-sectional and prospective studies in humans and experimental studies. Epidemiological studies show a statistical link between exposure to pesticides, polychlorinated bisphenyls, bisphenol A, phthalates, aromatic polycyclic hydrocarbides, or dioxins and insulin resistance.

## 1. Background

Endocrine-disrupting chemicals (EDCs) are a class of chemicals that could increase the risk of disease across the lifespan by interfering with the homeostasis or with the action of endogenous hormones or with other signaling chemicals of the endocrine system [1]; Figure 1 depicts the key features of EDCs. Infants and children may have higher exposure to some EDCs than adults due to differences in diet, behavior, physiology, anatomy, and pharmacokinetics [2]. In fact, infants and children may be more sensitive to the effects of EDCs than adults for several reasons. Differences in toxicokinetics can result in higher circulation or tissue concentrations of an EDC for an administered dose. For instance, the fetus has lower levels of several cytochrome P450 enzymes that metabolize environmental chemicals and pharmaceuticals compared to adults [3,4]. There are many time-dependent and synchronized processes that are programmed during early development, which could increase the risk of childhood disease if disrupted.

Early-life EDC exposures may promote childhood obesity, cardiometabolic impairment, and liver dysfunction by perturbing the neuroendocrine system [5,6,7,8]. These perturbations may lead to a quick early life weight gain and excess adipose mass. Rapid growth and excess adiposity lead to increased circulating levels of free fatty acids, causing a cascade of metabolic changes that result in altered glucose metabolism [9,10,11] with consequent increased pancreatic insulin secretion and resistance.

Insulin resistance (IR) is a decreased tissue response to insulin-mediated cellular actions [12], with a decreased ability of insulin to stimulate the use of glucose and suppress hepatic glucose and output. Moreover, it reckons for resistance to insulin action on protein and lipid metabolism and on vascular endothelial function and gene expression [13].

IR is a complex condition with genetic and environmental factors implicated in its etiology [13]. Several environmental factors can influence insulin sensitivity: obesity, ethnicity, sex, perinatal factors, puberty, sedentary lifestyle, and diet [13]. IR is closely related to obesity; excess of adipose tissue is one of the most important causes of IR. It is well-known that IR is a shared pathological condition associated with several dysmetabolic statuses including obesity and type 2 diabetes (T2D), dyslipidemia, atherosclerosis, polycystic ovarian syndrome (PCOS), and non-alcoholic fatty liver disease (NAFLD) [14].

The etiology of these conditions is multifactorial; although the genetic background of the individual and lifestyle play a key pathogenetic role, increasingly robust scientific evidence shows that endocrine disruptors (EDCs) clearly contribute to the development of obesity and changes in carbohydrate and lipid metabolism in both humans and animals [15].

Emerging shreds of evidence regarding the role of EDCs in the development of obesity and metabolic disturbances have been postulated in the last few years. The Parma consensus statement produced in May 2014 suggested the hypothesis that many of these pollutants may induce metabolic abnormalities such as IR and obesity in humans and animals.

Endocrine disruptors contribute to the manifestation of the metabolic syndrome through inflammatory processes via cytokines/adipokines, producing the effects of metabolic imbalance [16]. EDCs can lead to IR both through a mechanism that induces obesity and direct action on pancreatic beta cells. Environmental pollutants can affect multiple aspects of β-cell physiology, including β-cell function and survival, insulin release, and glucose provision [17]. BPA not only can cause weight gain but also lead to glucose intolerance, T2D, and fatty liver in mice [18].

This review will focus on EDCs for which there is widespread general population exposure.

**Figure 1 biomedicines-08-00137-f001:**
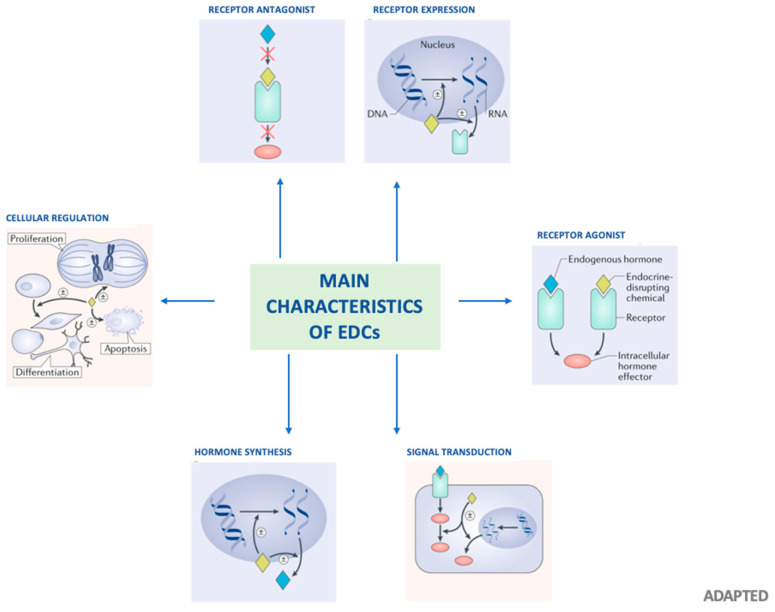
Main features of EDCs, Adapted by Michele A La Merill et al., Nature Reviews 2020 [19]. Arrows identify the main characteristics of EDCs. The ± symbol indicates that an EDC can increase or decrease processes and effects. EDC can interact with or activate hormone receptors, antagonize hormone receptors, alter hormone receptor expression. Furthermore, EDC can alter the fate of hormone-producing or hormone-responsive cells. Alter hormone synthesis and cellular regulation.

### 1.1. Phthalates

Phthalates are the esters of 1,2-dibenzene dicarboxylic acid; their general structure is outlined in Figure 2. Phthalates are a class of EDCs used in a multitude of consumer products, including personal care items, medications, and plastics. Biomonitoring studies from all over the world indicate that there are universal phthalate exposure among pregnant women, infants, and children [20,21,22]. Phthalate exposure occurs through ingestion, inhalation, or dermal absorption [23,24,25,26]. It has been shown that phthalates can cross the placenta, resulting in exposure to the fetus [27].

After ingestion or inhalation, the phthalates are rapidly hydrolyzed to their respective mono-ester metabolites [28]. Low molecular weight phthalates (di-ethyl phthalate [DEP], di-n-butyl phthalate, and di-iso-butyl phthalate) are excreted in the urine as glucuronide or sulfate-conjugated hydrolytic monoesters, while mono-2-ethylhexyl phthalate, the hydrolytic metabolite of di-2-ethylhexyl phthalate (DEHP), undergoes additional enzymatic oxidation before being conjugated and excreted. Even though phthalates do not persist in the body and have short biological half-lives (<24 h), there is a repeated exposure. Phthalate exposure is assessed using urine biospecimens since phthalates are predominately excreted in the urine and blood levels, which are considerably lower, may be subject to exogenous contamination during sample collection, storage, or processing [29]. Accurate phthalate exposure assessment necessitates the collection and analysis of multiple urine samples [30]. Phthalates may interfere with the action or metabolism of androgens, thyroid hormones, and glucocorticoids.

### 1.2. Perfluoroalkyl Substances

Perfluoroalkyl substances (PFAS) are a class of man-made fluorinated chemicals used in stain/water resistant coatings for textiles, non-stick cookware, food container coatings, fire-fighting foam, floor polish, and industrial surfactants [31]. PFAS are found in a wide range of consumer products (Figure 3).

PFAS are hallmarked by the presence of multiple fluorine atoms attached to an alkyl chain.

PFAS, due to the strong C-F chemical bond, are extremely resistant to thermal, chemical, and biological degradation, which results in bioaccumulation and persistence in human tissues for years [32]. PFAS have long biological half-lives in humans, ranging from 3.8 to 7.3 years, and some PFAS (perfluorooctanoic acid, perfluorooctane sulfonate, perfluorononanoic acid, and perfluorohexane sulfonate) can cross the placenta [11,21,33,34,35].

### 1.3. BPA

Bisphenol A (BPA) (Figure 4) is an organic synthetic compound with the chemical formula (CH_3_)_2_C(C_6_H_4_OH)_2_ belonging to the group of diphenylmethane derivatives and bisphenols, with two hydroxyphenyl groups.

BPA is used to produce polycarbonate plastics and resins that are used in a wide range of consumer products. Oral ingestion is the predominant exposure route since BPA can leach into food and beverage containers; however, dermal absorption and inhalation may be additional routes of exposure among persons working with BPA-containing receipts [36,37,38]. BPA is excreted in the urine as glucuronide/sulfate conjugates, does not persist in the body, and has an estimated biological half-life of six hours [39]. BPA exposure is assessed by measuring urine concentrations of free and conjugated BPA due to the fact that BPA is almost exclusively excreted in the urine [29]. Biomonitoring studies worldwide indicate nearby universal BPA exposure among pregnant women, infants, and children [21,40,41]. Current safety assessments by the preponderance of international regulatory agencies (Healt Canada, 2012; EFSA, 2015; FDA, 2014) conclude that BPA at current exposure levels does not jeopardize humans via dietary exposure at any life stage; nevertheless, studies showed that BPA is likely to be a human health hazard [42].

### 1.4. Triclosan

Triclosan is a chlorinated aromatic compound that has functional groups representative of both ethers and phenols (Figure 5).

Triclosan is an antimicrobial chemical that disrupts bacterial lipid synthesis and cell membrane integrity and is used in numerous consumer products [43]. Exposure is mainly through oral and dermal routes [43]. Triclosan is not persistent, has a biological half-life of fewer than 24 h, and is predominately excreted in the urine as a glucuronide or sulfate conjugate [44]. Triclosan exposure is measured using urine biospecimens for the same reasons that BPA and phthalates are measured using urine biospecimens [28]. Biomonitoring studies indicate nearly universal triclosan exposure among pregnant women and children [33,45,46].

## 2. Obesogenic Hypotesis

Several EDCs have been related to the development of obesity. Grün and Blumberg used the term “obesogenic” related to EDCS for the first time in 2006 [47]. Several studies have shown how EDCs may alter energy homeostasis both in cellular and animal models as well as in humans. EDCs can act with different mechanisms: increase in number and size of adipocytes, impairment of endocrine regulation of adipose tissue and adipocytokine production, reduction of basal metabolic rate, and changes in the regulation of appetite and satiety. These effects are due to molecular actions of EDCs on cellular function via interaction with steroid receptors and nuclear transcription factors, compromising of endocrine signaling transduction and epigenetic mechanisms.

EDCs can affect both fetal growth and subsequent action over the years, and some EDCs also act on the prenatal period [48].

Braun examined the relationship between early-life exposure to EDCs and childhood obesity [21].

The association between early-life phthalate exposure and childhood obesity does not support the hypothetical obesogenic role of phthalate [20,49,50]. As far as BPA concern, literature is ambiguous about the obesogenic effects of early-life exposure [41,51,52,53,54,55].

Prenatal exposure to triclosan has been described mainly by Xue J et al. and by Li S et al. [56,57]; there is insufficient evidence to assess if early-life triclosan exposure is associated to obesity in children. The association between prenatal exposure to PFAS and the development of obesity has been observed in human and animal model studies [35]. PFAS exposure can lead to altered fetal growth, which in turn may cause obesity and cardiometabolic disorder; PFAS exposure is associated with abnormalities in infant and child growth [22,58] and with increased adiposity in children and adults [22,59,60].

In vivo and in vitro models mainly studied EDCs interaction with peroxisome proliferator-activated receptor (PPAR) γ and retinoid X receptor (RXR), anti-androgenic/xenoestrogenic action and interaction with the hypothalamic–pituitary–thyroid (HPT) axis [16]. PPARγ is a nuclear transcription factor highly expressed in adipose tissues during adipogenesis [61]; PPARy is considered the principal regulator of adipogenesis [62,63].

Upon activation, PPARc heterodimerizes with the retinoid X receptor (RXR) and regulates the expression of genes involved in adipogenesis and adipocyte differentiation from stem cells [64,65].

In vivo and in vitro models have underlined the capacity of EDCs to interact with PPARγ inducing adipogenesis and lipid storage in adipose tissue [47,66,67,68]. The interaction of EDCs with PPAR-RXR may contribute to the development of the pro-inflammatory state typical of metabolic syndrome and obesity [69,70].

EDCs may perform anti-androgenic and xenoestrogenic actions; androgens and estrogens are involved in the regulation of lipid and glucose metabolism and in the regulation of adipose tissue [71,72]. As a result, EDCs can exert their obesogenic action by altering sex hormones receptor pathways, inhibiting the androgen receptor pathway, enhancing the estrogen pathway, or reducing androgen conversion through the upregulation of the aromatase enzyme [73,74,75].

In addition, EDCs may exert an obesogenic role by interfering with thyroid function. Thyroid hormones play a key role in the regulation of basal metabolism and energy expenditure [76]. The role of EDCs in the development of the metabolic disease may, therefore, be partly related to the disruption of the hypothalamic–pituitary–thyroid axis [16,77,78].

Other EDC actions that have a possible implication in the development of obesity have been pointed out recently [16]. EDCs have been shown to disrupt the function of metabolic physiological contrast against oxidative stress [79], enhancing the inflammatory condition of obese subjects.

Finally, EDCs can contribute to the development of obesity by affecting intestinal microbiota [80]. Both the gastrointestinal tract and its microbiota are exposed to EDCs through the diet. EDCs’ dietary exposure has been shown to alter the composition of the microbiota.

These changes are linked to disorders in the immune homeostasis of the host intestine with subsequent changes in cytokine production and metabolism of liver lipids and glucose [81].

The main mechanisms underlying the obesogenic theory is presented in Table 1.

## 3. Diabetogenic Hypothesis

In recent years, there has been a relevant increase in the production of synthetic chemicals; EDCs can be considered as diabetogenic compounds, independently of their impact on adipose tissue metabolism [83].

Diabetogenic chemicals can exert their action either by impairing insulin production in pancreatic beta cells or by disturbing insulin sensitivity in peripheral tissues.

EDC actions on pancreatic function can occur through several mechanisms; for example, TBT reduces beta cell mass and enhances β cell apoptosis [84]; phthalates reduce β cell insulin content [85]; and environmentally relevant doses of BPA (1 nM) stimulated glucose-induced insulin secretion in human islets [86].

In animal models, BPA alters hepatic glucose sensing, impairing glucokinase (GCK) specific activity [87].

A meta-analysis of the cross-sectional and prospective studies, published in 2016 by Song Y et al., showed significant relationships between levels of dioxins, PCBs, organochloride pesticides and BPA, and prevalent diabetes; meanwhile the relationship for phthalates was of borderline significance [88]. The relationship between PFNA and diabetes was confirmed in a cross-sectional study by Lind et al. in adults [89]. Prospective evidence exists for associations between background exposures to PCBs and organochlorine pesticides and incident diabetes; cross-sectional evidence exists for relationships between dioxins and BPA and prevalent diabetes.

Insulin resistance can be provoked by various molecular patterns involved in obesogenic conditions (Figure 6) [79]. EDCs can interfere with hormones and other factors such as leptin, adiponectin, resistin, and adipsin.

TNF-alfa decreases insulin sensitivity by restricting glucose transporter type 4 (GLUT4) function [90]. Leptin regulates intracellular lipid levels in hepatic and β pancreatic cells, increasing insulin sensitivity [91]. As for the proteins involved in cellular signaling, resistin and vifastin are involved in the regulation of insulin secretion and insulin sensitivity.

Resistin is a member of cysteine-rich protein which appears to increase in T2D, obesity, and insulin resistance; resistin promotes insulin sensitivity through TNF-alfa and IL-6 activation [92]. Visfatin, also known as pre-β cell colony enhancing factor (PBEF), performs several functions including maturation of β cells and hypoglycemic effect derived from decrease of glucose release from liver. Vifastin level are increased in obesity and T2D; vifastin promotes adipocytes maturation and mimics insulin binding to its receptor at a site different from that of insulin [93]. Adipsin is an adipokine with a beneficial role in maintaining β function. Gomez-Banoy et al. showed that higher concentrations of circulating adipsin are associated with a significantly lower risk of developing future diabetes [94].

Adipsin-C3/B interaction inhibits lipolysis and glucose transportation [95].

In addition, insulin resistance is caused by inhibition of GLUT4 due to excessive expression of RBP-4 (retinol binding protein-4) in abnormal adipose tissue [96]. As insulin resistance occurs, there is an increase in fasting glucose and this metabolic state induces hyperinsulinemia, which simulates transcription factors in the liver, driving hypertriglyceridemia and hepatic steatosis [97].

Hyperglycemia with reduced insulin levels was found in females exposed to diethylhexyl phthalate throughout the gestation/perinatal period [85]. Bodin et al. showed an increased severity of insulitis in rats exposed to BPA during the perinatal period [98]. EDCs induce genome alterations in pregnancy or early life and enchain in a decreased expression of pancreatic/duodenal homeobox 1 transcription factor gene (PDX-1)/increase of T2D [99], suggesting that in utero exposure to impaired-nutrition is a risk for obesity and diabetes progression in adulthood. EDCs confine the stock of essential metabolic substrates to the fetus and cause intrauterine growth retardation, presenting as fetal starvation and the metabolic basis that triggers diabetes progression in PDX-1. Prenatal and early-life exposure to BPA, perfluorinated compounds, PCBs, and dioxins may negatively affect the development of the immune system, resulting in immune disorders such as type 1 diabetes mellitus [100,101]; hormonal and epigenetic alteration can be involved.

EDCs deregulate pancreatic islet beta-cell function, development of peripheral IR, insulin production, beta-cell mass (compensatory hyperplasia/hypertrophy of beta cells) and impaired insulin output, insulin signaling, and increasing cell apoptosis [79]. EDCs promote, by these ways, the onset of diabetes in obese insulin-resistant individuals with T2D [102]. For example, PCB induces pre-proinsulin expression via AHR activation and inhibition of transcription factor Nrf2a [103] and PFOA significantly increases the proinsulin/insulin ratio [88]. EDCs increase the risk of T2D through modulation of glucose metabolism [104]. Organochlorines (OCs) and PCBs act through mitochondrial dysfunction and endocrine-disrupting mechanisms [105], including PCBs effects on pancreatic beta-cell function [106] and OCs adiponectin release [107]. EDCs decrease GLP-1R (glucagon-like peptide 1 receptor); it increases release of pancreatic glucagon via hypothalamic receptors as a lack of satiety during eating [54].

EDCs play a key role in obesity-associated IR due to activation of the extracellular matrix receptor pathways in adipose tissue that constitute the cell microenvironment [108]. EDCs involved in extracellular matrix remodeling through its receptors such as integrins and CD44 contribute to inflammation, apoptosis, and angiogenesis in adipose tissue as well as in skeletal muscle and liver tissue [74]. Since excessive extracellular matrix deposition results in adipose tissue fibrosis overpassing angiogenesis capability in tissues, repressed expression of genes essential for adipose angiogenesis (e.g., VEGFa) appear to be mediated by activation of extracellular matrix receptor and HIF1a/VEGFa pathways [109]. EDCs can also be modulated by other signaling pathways [110], including EGF, IGF, integrins (fibronectin), phosphatidylinositol3-kinase/AKT, X-box binding protein 1 (XBP1), and second messengers cAMP and dopamine; they can also influence estrogen receptor (ER) transcriptional activity by targeting the receptor directly or by regulating co-regulators [111].

EDCs affect quantitative insulin secretion and immunity but also alter insulin-dependent mRNA stability.

Because insulin-like growth factor-binding protein gene (IGFBP-1) promoter regulates blood glucose levels, the specific upregulation of IGFBP-1 mRNA in human hepatocytes and HepG2 human hepatoma cells, even in the presence of insulin, might account for the disruptive effects of TCDD on glucose metabolism [112].

EDCs reduce insulin sensitivity acting on insulin targets, particularly in the liver.

Environmental contaminants can negatively affect multiple aspects of β-cell physiology; for example, dioxin exposure, mainly TCDD, was one of the first to be linked with metabolic alterations in multiple experimental studies. TCDD decreases glucose uptake in pancreas and impair insulin secretion [113] and leads to the consumption of the cellular insulin reservoir [114], suggesting that insulin deficiency may ensue after sustained exposure to this compound.

Diethylstilbestrol (DES) exposure can lead to a deleterious fetal nutritional environment, leading to intrauterine growth retardation and influencing the later occurrence of insulin resistance [82,115]. At the cellular level, tributyltin can reduce beta-cell mass and induce beta-cell apoptosis [84].

Other EDCs can disturb pancreatic function: oral administration of TBT was shown to inhibit the proliferation and induce the apoptosis of islet cells via multiple pathways, causing a decrease of relative islet area in the animals treated for 60 days, which could result in dysregulation of glucose homeostasis [84]. Arsenic has also been correlated with the onset of insulin resistance downregulating insulin gene expression [116] and interfering with insulin granule exocytosis through calpain-10-mediated proteolysis and activation of SNAP-25 [117]. Several studies have suggested adverse endocrine disruptive effects of BPA on the endocrine system.

BPA has also been investigated as a disrupter of pancreatic beta cells. Animal studies show that pregnant mice treated with BPA during gestation, at environmentally relevant doses, exhibit profound glucose intolerance and altered insulin sensitivity, and become overweight several months after delivery, mainly through impairments in beta-cell function and mass [118]. In vivo experiments suggest that BPA exposure increases insulin release and glucose-stimulated insulin secretion in an estrogen receptor-a (ERa) dependent mode [119]. Sex steroids exert important effects on metabolic target tissues, including the pancreas, checking β-cell insulin secretion in both cGMP-dependent and independent pathways. Several EDCs, including BPA, arsenic, and DEHP, can disrupt β-cell function, promoting oxidative stress [120]. Oxidative stress can significantly compromise β-cell function, as pancreatic β cells are innately more sensitive. For instance, long-term exposure to BPA triggered spontaneous insulitis in non-obese diabetic (NOD) mice, a model of immune-mediated diabetes, suggesting that BPA can accelerate the exhaustion of β-cell reserve via immune modulations in pancreatic islets. As it becomes obvious, the immunomodulatory effects of BPA in this animal model suggest that EDCs might also possibly contribute to the increasing T1DM prevalence.

In 2019, Camacho et al. reported the data from the guideline-compliant two-year toxicology study conducted as part of the Consortium Linking Academic and Regulatory Insights on Bisphenol A Toxicity (CLARITY-BPA) during any early developmental stage in rodents.

The authors investigated several issues; as far as the metabolic aspect is concerned, mean body weights of females in the 250 μg BPA/kg bw/day dose group were significantly higher by 16−18% than those of the vehicle control group [42].

Furthermore, they found a possible relationship between the increased incidences of lesions in the female reproductive tract and the male pituitary gland and exposure to the 25,000 μg BPA/kg bw/day dose level.

However, the current evidence has not conclusively established an association between EDCs and metabolic abnormalities, especially in children, making more perspective studies mandatory.

## 4. Conclusions

In the last decade, there has been a growing interest in the possible health threat determined by EDCs, which are present in our environment, food, and consumer products that may interfere with hormone biosynthesis, metabolism or action; consequently, they might result in a deviation from normal homeostatic control or reproduction. This effect could be particularly important in fetuses, newborns, and children. In fact, exposure to isolated EDCs or EDCs mixtures, even at low doses, especially during crucial window time periods, can impact the endocrine system through several mechanisms.

There is a growing body of evidence showing that exposure to EDCs may adversely impact the health of adults and children through altered endocrine function; a body of evidence has shown that exposure to EDCs may be linked to endocrinopathies and obesity, but further studies are now needed to obtain more robust data on the correlation between EDCs and insulin resistance, especially in children.

## Figures and Tables

**Figure 2 biomedicines-08-00137-f002:**
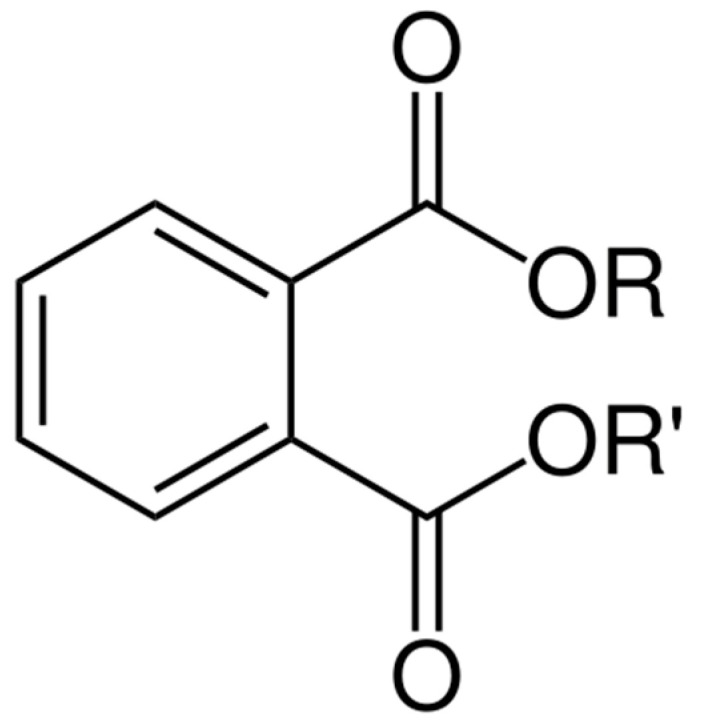
Chemical structure of phthalates (R: alkyl group).

**Figure 3 biomedicines-08-00137-f003:**
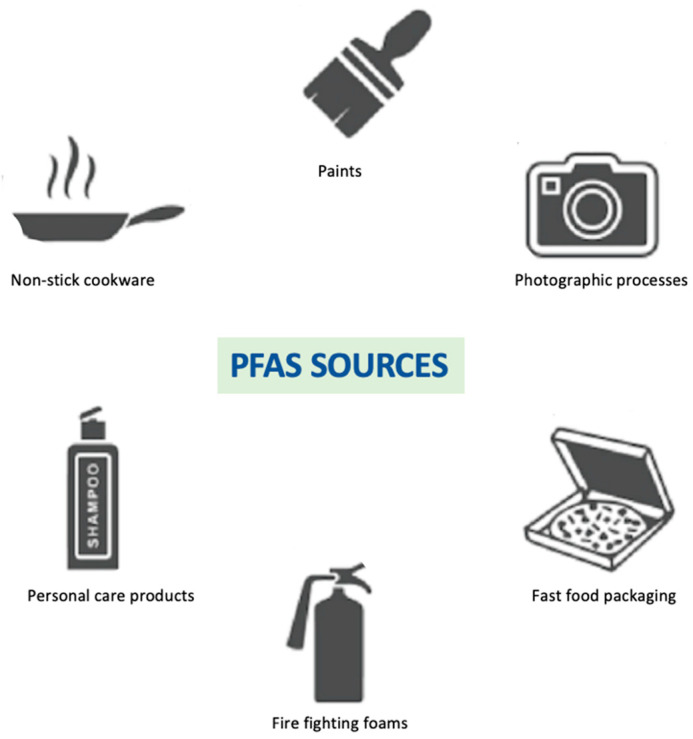
Main sources of PFAS.

**Figure 4 biomedicines-08-00137-f004:**
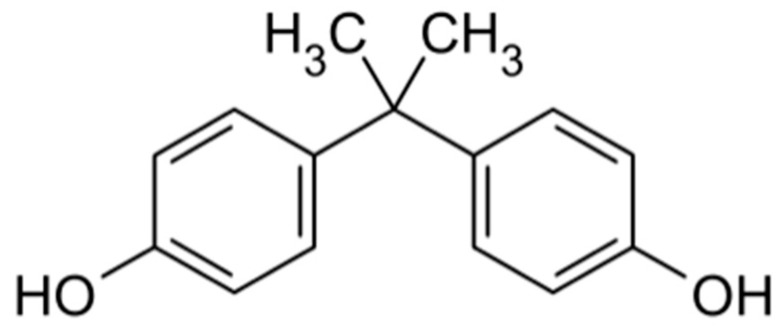
Chemical structure of BPA.

**Figure 5 biomedicines-08-00137-f005:**
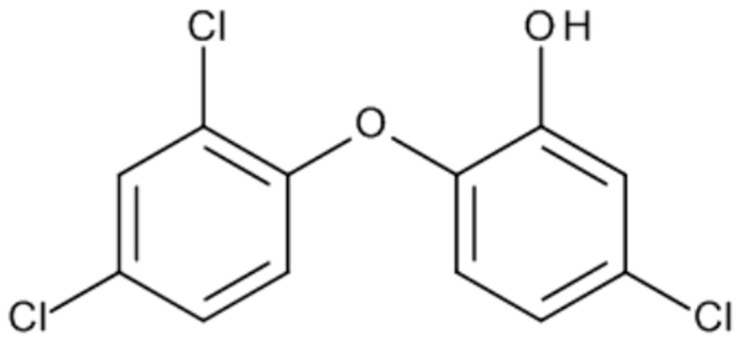
Chemical structure of Triclosan.

**Figure 6 biomedicines-08-00137-f006:**
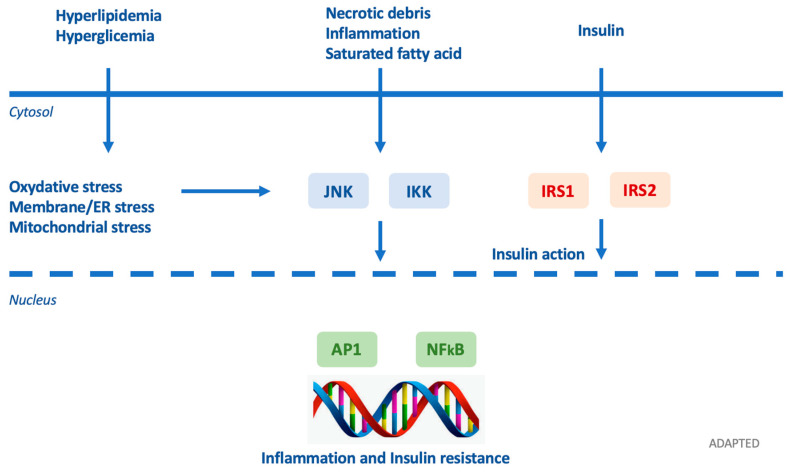
Adapted by Petrakie et al. Inflammatory signaling pathways link nutrient excess to insulin resistance. Cytoplasmic/nuclear responses via tyrosine phosphorylation of insulin receptor substrate (IRS)-1 and IRS-2 are activated by the presence of insulin at the cell surface. Nevertheless, insulin signaling is potentially inhibited by serine phosphorylation of these proteins by Jun N-terminal kinases (JNK) and inhibitor of nuclear factor κB (NF-κB) kinases (IKK). Various intra/extracellular sequelae of chronic nutrient excess activate these signaling pathways, linking overfeeding to insulin resistance. JNK and IKK activation triggers inflammatory cytokine production, activating JNK/IKK in an autocrine/paracrine manner, further reinforcing insulin resistance. ER: endoplasmic reticulum; AP-1: activator protein-1. Arrows identify the subsequentiality of events.

**Table 1 biomedicines-08-00137-t001:** Obesogenic hypotesis, adapted by Heindel et al. [82].

Mechanism	Results
PPAR and RXR Activation	adipogenesis induction stimulation of lipid storage in adipose tissue imbalance in adipocytokine production
Anti-androgenic action	agonistic activity on estrogen receptors antagonistic action on androgen receptors *aromatase upregulation*
Perturbation of the hypothalamic–pituitary–thyroid (*HPT*) *axis*	interference with thyroid hormone synthesis, release, transport and metabolism interference with the action of thyroid hormones on target tissues
Epigenetic actions	DNA metilation hystonic modifications microRNA

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
