# Peer review of "Endocrine-Disrupting Chemicals and Insulin Resistance in Children"

_biomedicines, 2020, doi:10.3390/biomedicines8060137_

Round 1
Reviewer 1 Report
Synopsis
This is a review of some of the literature available regarding the obesogenic and diabetogenic effects of environmental chemicals.
Critique
Extensive edits and revisions are needed to correct for English usage and grammar.
Extensive revisions are needed to increase the readability of the paper. For example, the statement on lines 40 – 42 is nonsensical and re-states the same idea multiple times.
Essential typographical errors need to be corrected. For example, Line 188 the authors refer to “Bodjn et al.” whereas the correct reference is “Bodin et al.”
Inaccurate or untrue statements such as …”BPA impairs insulin secretion” on line 165 need to be corrected. In reference # 84, Soriano et al., they report an insulinotropic effect of BPA. Meaning there was an increase in glucose-dependent insulin release when BPA was present. This is the opposite of what the authors of this paper stated in referencing the Soriano et al., study.
Publications showing conflicting effects of endocrine disrupting chemicals are not discussed especially those of BPA. For example on Line 224 the authors state BPA causes impaired beta-cell function while in the next sentence lines 244 and 246 they cite another study showing that BPA increases glucose stimulated insulin secretion. Lastly, the 2019 publication by Camacho et al., “A two-year toxicology study of bisphenol A (BPA) in Sprague-Dawley rats: CLARITY-BPA core study results” needs to be incorporated in a thoughtful discussion of the other relevant BPA studies.
Author Response
We thank the reviewer for precious suggestion.
We revised the manuscript and we added suggested reference.
Reviewer 2 Report
Review entitled “Endocrine-Disrupting Chemicals and insulin resistance in children” biomedicines-760180, by Rotondo and Chiarelli
This review is well written and structured that aims to describe the effects of different chemical compounds over insulin metabolism.
Personally, the Abstract is fine, because it summarizes the content of the full manuscript.
However, several questions need to be improved.
It is necessary to introduce a final section of Conclusions
In each section, there are several new paragraphs (even with a single sentence) that do not allow us to follow the meaning of these sections and these paragraphs are apparently disconnected from one to others.
It is necessary to review the use of abbreviations throughout the manuscript, even there are a box wioth some of them. Sometimes, these abbreviations are not presented in the text (e.g. BPA; PCBs, AHR, …) or the meaning of the abbreviation has sometimes been presented, but the full words are reused (e.g. type 2 diabetes).
About the tables, perhaps is better to use the term “adapted” than “modified”.
It is necessary to explain figure 1, since several of the terms used are not explained in the text (necrotic debris, oxidative stress, JNK, IKK, ...) and a complete figure legend is also necessary.
Regarding figures, extra figures will improve the manuscript and will facilitate the reading compression: showing similar chemical structures between EDCs and some hormones, another explaining the EDCs target tissues, biodisponibility, EDCs sources, differences and similarities between adult and child EDCs exposition, …
I
Minor points:
It is necessary to introduce a space between words in the lines 177 and 202.
Revise the reference of the 171 line.
Line 84: which is the meaning of 177 or it is a mistake?
Author Response
We are relly grateful for the reviewer's efforts to improve our manuscript.
We revised the manuscript and we added figures.
Round 2
Reviewer 1 Report
This is a review of endocrine disrupting chemicals that is a highly revised version of the original submission; especially section "3. Diabetogenic Hypothesis". The authors have done an excellent job of incorporating reviewers' comments to make this manuscript more interesting to prospective readers and a valuable contribution to the literature overall.
Author Response
We are really grateful for the reviewer’s comment.

Reviewer 2 Report
The authors have fulfilled the reviewer recommendations and the manuscript have clearly improved.
Please, revise the reference section because it appears with a doubled numeration and which is not consistent from reference 120.
Author Response
We thank the reviewer for comment. We revised the reference section.
